# Adaptive Radiation from a Chromosomal Perspective: Evidence of Chromosome Set Stability in Cichlid Fishes (Cichlidae: Teleostei) from the Barombi Mbo Lake, Cameroon

**DOI:** 10.3390/ijms20204994

**Published:** 2019-10-09

**Authors:** Zuzana Majtánová, Adrian Indermaur, Arnold Roger Bitja Nyom, Petr Ráb, Zuzana Musilova

**Affiliations:** 1Laboratory of Fish Genetics, Institute of Animal Physiology and Genetics, Czech Academy of Sciences, 27721 Liběchov, Czech Republic; rab@iapg.cas.cz; 2Zoological Institute, University of Basel, 4051 Basel, Switzerland; a.indermaur@unibas.ch; 3Department of Biological Sciences, University of Ngaoundéré, Ngaoundéré P.O. Box 454, Cameroon; bitja.nyom_arnold@ymail.com; 4Department of Management of Fisheries and Aquatic Ecosystems, University of Douala, Douala P.O. Box 2701, Cameroon; 5Department of Zoology, Faculty of Science, Charles University in Prague, 12844 Prague, Czech Republic

**Keywords:** karyotype, rDNA, chromosome banding, FISH, adaptive radiation, cytotaxonomy, chromosome stasis, African endemic fishes

## Abstract

Cichlid fishes are the subject of scientific interest because of their rapid adaptive radiation, resulting in extensive ecological and taxonomic diversity. In this study, we examined 11 morphologically distinct cichlid species endemic to Barombi Mbo, the largest crater lake in western Cameroon, namely *Konia eisentrauti*, *Konia dikume*, *Myaka myaka*, *Pungu maclareni*, *Sarotherodon steinbachi*, *Sarotherodon lohbergeri*, *Sarotherodon linnellii*, *Sarotherodon caroli*, *Stomatepia mariae*, *Stomatepia pindu*, and *Stomatepia mongo*. These species supposedly evolved via sympatric ecological speciation from a common ancestor, which colonized the lake no earlier than one million years ago. Here we present the first comparative cytogenetic analysis of cichlid species from Barombi Mbo Lake using both conventional (Giemsa staining, C-banding, and CMA_3_/DAPI staining) and molecular (fluorescence in situ hybridization with telomeric, 5S, and 28S rDNA probes) methods. We observed stability on both macro and micro-chromosomal levels. The diploid chromosome number was 2n = 44, and the karyotype was invariably composed of three pairs of meta/submetacentric and 19 pairs of subtelo/acrocentric chromosomes in all analysed species, with the same numbers of rDNA clusters and distribution of heterochromatin. The results suggest the evolutionary stability of chromosomal set; therefore, the large-scale chromosomal rearrangements seem to be unlikely associated with the sympatric speciation in Barombi Mbo.

## 1. Introduction

The intralacustrine speciation of fish species flocks has attracted the attention of biologists since the beginning of modern ichthyology, and lead to the discovery of rich and speciose lacustrine fish faunas, such as cichlids in African lakes [1,2], cottid sculpins in Lake Baikal [3,4], cyprinids of Lake Lanao in the Philippines [5], barbs of Tana Lake in Ethiopia [6], or coregonids in the Great Lakes in North America or northern Europe [7,8,9]. Besides these spectacular lacustrine fish faunas in ancient lakes [10,11,12], some smaller-scale fish species flocks have been discovered, such as the telmatherid sail-fin silversides in Malili lakes in Sulawesi [13], homalopterid genus *Yunnanilus* in the Yunnan province of China [14], killifishes of the genus *Orestias* in the Andean Altiplano [15,16], and the cichlids of Barombi Mbo crater lake in Cameroon [17,18], among many others. The differentiation and speciation of these lacustrine fish faunas have been assessed from various perspectives, but cytogenetic tools have been used only in a small number of cases. Using this approach, Symonová et al. [19] analyzed deep-water cisco species *Coregonus fontanae* from Lake Stechlin, Germany and its differentiation from ubiquitous, shallow-water European Vendace, *Coregonus albula*. Similarly, Dion-Côté et al. [20,21] studied incipient speciation process within various forms of Lake Whitefish *Coregonus clupeaformis*. In spite of the fact that cichlids are a textbook example and the focus of numerous studies on intralacustrine speciation, only a handful of studies aim to address karyotype evolution in African cichlids (e.g., [22,23]), and the comprehensive attempts to apply cytogenetic methods to understand details of the speciation process are still largely lacking.

The family Cichlidae includes more than 3000 species, comprising one of the most species-rich clades of freshwater euteleosts [24], distributed from Central and South America, across Africa to Madagascar and southern India [25,26]. The African (Pseudocrenilabrinae) and Neotropical (Cichlinae) cichlids are reciprocally monophyletic and sister taxa [27], with the divergence age estimated to 81–103 Mya (95% HPD 74–123 Mya [28,29]). The exceptional diversity and propensity to generate adaptive radiations have made cichlid fishes one of the most important vertebrate model systems for evolutionary biology research [2,25,30]. One such example of adaptive radiation is represented by a species flock from the Barombi Mbo Lake, the largest and the deepest volcanic crater lake in Cameroon. Despite the size of the lake not exceeding 7 km^2^ [31], it harbors an endemic monophyletic radiation of 11 cichlid species. This monophyletic species flock has most likely radiated in situ, via sympatric ecological speciation [32,33,34] from the common ancestor, probably *Sarotherodon galilaeus* (Oreochromini), which colonized the lake no earlier than one million years ago [17,18,35] when the last substantial geological activity was detected [36,37]. Cichlids from the Barombi Mbo Lake are classified in five genera, four of which are endemic to the lake (i.e., *Konia*, *Myaka*, *Pungu*, and *Stomatepia*), and one genus (*Sarotherodon*) is widely used also for other African mostly riverine cichlids [32,38]. The Barombi Mbo cichlid fishes are emerging as a model system for evolutionary biology, namely focusing on mechanisms of speciation, adaptive radiation, ecological diversification, or evolutionary history in general (e.g., [17,18,28,31,32,39,40]).

The intriguing question of sympatric speciation is how multiple species diversify in close geographic proximity without any physical barriers and with ongoing gene flow [41]. Karyotype differentiation and chromosomal rearrangements are one of the possible mechanisms able to cause postzygotic reproductive barriers, and therefore, may trigger or facilitate speciation (as postulated by [42,43] and reviewed in [44]). Such chromosomal speciation is broadly evidenced, e.g., from mammals (shrews [45], wallabies [46]), plants [47], or butterflies [48], however, cases in teleost fishes are generally more rare, such as in the swamp eels [49] or in chromosomal races in trahiras (Characiformes [50]). Smaller-scale chromosomal rearrangements not noticeable on the karyotype level (i.e., number and morphology of chromosomes), but detectable by the advanced techniques (such as rDNA staining) have been evidenced and hypothesized to be associated with speciation, for example, in *Coregonus* [19]. Similarly, in African cichlids, recent studies focused on the chromosome-level whole-genome assemblies of two species identified mostly intrachromosomal rearrangements [51], yet no clear case of speciation associated with the karyotype incompatibilities has been reported so far in cichlids. Here we present the first comparative cytogenetic research focused on the Barombi Mbo cichlid fishes in order to test if large-scale chromosomal rearrangements might have contributed to the speciation in the lake.

So far, karyotypes of almost 200 species of cichlids have been determined [22,23,52,53,54,55,56,57]. Diploid chromosome numbers range from 2n = 32 to 2n = 60, more than half of studied species possess karyotypes with 2n = 48 [52], which is the modal number for the Neotropical lineage, whereas African cichlids possess predominantly 2n = 44 (Figure 1, [23,52]). Even though cytogenetic data are known for a number of cichlid species, it still covers less than 10% of cichlid taxonomic diversity. In this study, we investigated karyotype differentiation in all 11 representatives of the Barombi Mbo Lake cichlid species flock to uncover trends in their chromosome and genome organization. Using molecular cytogenetic methods, we examined karyotypes and reconstructed the ideogram for the typical Barombi Mbo cichlid genome. We further identified the numbers and positions of minor and major rDNA genes on the chromosomes, and visualized the distribution of heterochromatic blocks and telomeric sequences. The main goal of this study was to address whether the sympatric speciation of this species flock may have been associated with karyotype or interchromosomal rearrangements, or finer-scale intrachromosomal variability, such as in the number of rDNA regions.

## 2. Results

### 2.1. Karyotypes

Karyotypes of all 11 species invariably possessed the diploid number, 2n = 44 (Figure 2, Table 1) and fundamental number, NF = 50. The general Barombi Mbo cichlid karyotype, identical for all species, consisted of three pairs of submetacentric chromosomes (sm) and 19 pairs of subtelocentric-acrocentric (st/a) chromosomes. Karyotypes of all studied species also possessed a large subtelocentric chromosome pair, a characteristic marker for the karyotype of the model species Nile tilapia (*Oreochromis niloticus* [58]). Results of the karyological analysis together with results of the comparative cytogenetic analyses, respectively, are summarized in the ideogram representing the typical Barombi Mbo cichlid karyotype (Figure 3).

### 2.2. Telomere Mapping

We applied the FISH telomeric mapping method to test for signs of any putative chromosomal rearrangements. By using FISH with the conserved vertebrate telomeric repeat (TTAGGG)_n_, we detected signals only at the termini of all chromosomes (Figure 4 and Figure 5). We did not detect any interstitial telomeric sequences (ITSs) in chromosomes of any of the examined species.

### 2.3. C-Banding

We applied the C-banding technique to detect accumulations of heterochromatin throughout the chromosomes. This method revealed similar distribution patterns of constitutive heterochromatin blocks among all studied species. In all studied species, we observed C-positive bands only in the pericentromeric regions of all chromosomes. No other large accumulations of heterochromatin were observed in the studied species. (Figure 4 and Figure 5).

### 2.4. CMA_3_/DAPI Staining

Reversed fluorescence staining (CMA_3_/DAPI) revealed homogeneous patterns across chromosomes and extremely GC-rich signals were found in the pericentromeric regions of multiple chromosomes. Species differed slightly in numbers of these GC-rich signals (Figure 4 and Figure 5). The highest number of the signals (18) was observed in the chromosomes of *Sarotherodon linnellii* and *Konia eisentrauti*, whereas the lowest number (14) was observed in *Pungu maclareni*. All other studied species possessed the GC-rich signals in the pericentromeric region of 16 chromosomes. 

### 2.5. Fluorescence In Situ Hybridization with rDNA Genes

The FISH (fluorescence in situ hybridization) experiments with rDNA probes showed the same numbers of clusters in the genomes of the nine studied species: three pairs of 28S rDNA hybridization signals and four pairs of 5S rDNA hybridization signals. The 28S rDNA probe hybridized at the pericentromeric region of two middle-sized acrocentric chromosomal pairs and at an interstitial position of one large acrocentric chromosomal pair. As for the 5S rDNA, two signals were located at the interstitial position of the largest pair, and six at the pericentromeric region of other three st/a chromosome pairs. One of the acrocentric chromosome pairs possessed both 28S and 5S rDNA loci (Figure 4 and Figure 5).

## 3. Discussion

Cichlids are one of the most speciose fish families in freshwaters worldwide, with about 3000 recognized species [24], of which about 2000 evolved in the adaptive radiation flocks of East African great lakes, namely, Lake Malawi, Lake Victoria, and Lake Tanganyika [2]. These species flocks have provided a model system for the study of evolution for decades [2,25,26,59]. Much smaller examples of adaptive radiation in cichlids have evolved in the crater lakes of Cameroon, such as Barombi Mbo, which hosts an endemic species flock of 11 cichlid species [32]. This flock is evolutionary quite recent, sharing a most recent common ancestor with the riverine species *Sarotherodon galilaeus* between 1 and 2.5 million years ago [28,35]. Most of the species in the lake have evolved probably by adaptive ecological speciation [33,34] triggered by the ecological differentiation. For example, deep-water specialists have evolved twice independently within the lake (*Konia dikume*, *Myaka myaka*; [34,60,61]), a mechanism also recently described from the onset of speciation in Massoko crater lake cichlids [62]. Further, different trophic strategies have been established within the flock, which includes predators, planktivores, insectivores, pure herbivores, and spongivores [33,34,60,61]. Ecological speciation in sympatry is a well understood evolutionary mechanism for example in sticklebacks (e.g., [63]) and in Lake Malawi cichlids [64]. Additionally, differential seasonality in spawning, as observed in one species (*Myaka myaka*) [32], is another potential mechanism for achieving sympatric speciation, as observed in arctic chars [65] and coregonids [19]. Alternatively, chromosomal rearrangements and subsequent reproductive isolation are also known mechanisms, causing speciation for example in guppies [66], whitefishes [20], recently happening in swamp eels [49] and possibly also beginning in the trahiras [50]. In this study, we tested the hypothesis of chromosomal differentiation within the sympatric species flock, and we found that the karyotypes of all eleven species within the Barombi Mbo species flock are very stable and similar to each other. This is comparable to the findings in the lake whitefish (*Coregonus clupeaformis*) system [20], where species retained the same karyotypes, but hybrid incompatibilities led to a dramatic reduction in embryonic survival in first- and second-generation hybrids. In this study, we did not detect any dynamics of chromosomal evolution that could have contributed to or triggered speciation within the Barombi Mbo species flock. 

Despite numerous genetic studies on lake cichlids, advanced karyotypic evolution has been studied only scarcely (e.g., [22,23]), and virtually nothing is known about karyotypic differentiation in the context of cichlid speciation. Karyotype data for cichlid fishes has been broadly published and almost 200 cichlid species have been cytogenetically analyzed [22,23,52,53,54,55,56,57]. Nevertheless, advanced molecular cytogenetic approaches have only been performed on a minority of the studied species. A lot of research attention has been focused on the Nile tilapia (*Oreochromis niloticus*), a model species closely related to the Barombi Mbo cichlids [23,67,68,69] or comparison with other African cichlids, recently including genomic data [51]. Both conventional and advanced molecular cytogenetic methods have been used to characterize genome evolution in cichlids (e.g., [70]). The ancestral diploid number of chromosomes (2n) for cichlids remain unclear (Figure 1), yet the modal 2n is different for different phylogenetic lineages. African cichlids have a modal 2n = 44 (but unclear ancestral 2n), whereas the Neotropical cichlids 2n = 48 [52] (and similarly, the ancestral 2n = 48; Figure 1). The chromosomal data published so far for the Pseudocrenilabrinae clade (= African cichlids) focuses mostly on the description of chromosome morphology and mapping of repetitive sequences [23,52]. Interestingly, B chromosomes have been identified in several species from lakes Victoria and Malawi in East Africa making the dynamics of chromosomal evolution even more complex [23,71,72]. No B chromosomes have been detected in this study or in Nile tilapia.

In this study, we provided cytogenetic analyses and description of chromosomal stability among the cichlid species of Barombi Mbo crater lake. All Barombi Mbo species are characterized by 2n = 44, the modal number of African cichlids [23], and the most prevalent number for the oreochromine lineage (Figure 1). Karyotypes of the Barombi Mbo cichlids are represented by three submetacentric chromosome pairs and 19 st/a chromosome pairs, including the large pair of subtelocentric chromosomes (Figure 2) characteristic also for karyotypes of many African cichlids, namely Nile tilapia (*O. niloticus* [58]). Three pairs of submetacentric chromosomes observed in the karyotypes of all Barombi Mbo species (Figure 2) is more than that in Nile tilapia with only one pair [23,68] (but see [67] who has also recognized three pairs); but lower than in, *S. galilaeus* (6 m/sm [73]) the closest related species to the Barombi Mbo cichlids [17]. This suggests that chromosomal rearrangements, such as centromeric shifts, possibly occurred in the ancestors of the flock to a certain extent. Previously, one representative from this species flock, *St. pindu*, was karyotyped [22], describing the same 2n = 44; however, the karyotype description itself differs slightly, possibly due to the different scoring of the (sub)metacentric and (sub)telocentric chromosomes. Similarly, even for the aforementioned model species, Nile tilapia, similar variation in scoring of sm vs st chromosomes is known (e.g., [23,68]—1 sm pair, [67]—3 sm pairs). We therefore did not aim to reach conclusions based on the comparison of different numbers of sm chromosomes, and rather focused on the chromosomal stability within the species flock.

Using advanced cytogenetic methods, we identified the similar number of clusters of 5S (four pairs of signals on four pairs of chromosomes) and 28S rDNA (three pairs of signals on three pairs of chromosomes) sequences among all nine studied species (Figure 3, Figure 4 and Figure 5). Additionally, one of the chromosomes possess both 5S and 28S rDNA signals (Figure 3). We observed the constitutive heterochromatin regions only in the pericentromeric regions of all chromosomes, the telomeric signals only in the telomeric regions, and no interstitial positions for the telomeric or heterochromatin signals were observed. All of the aforementioned analyses suggest that we can consider the karyotype of the Barombi Mbo species as very stable, with no signals suggesting recent chromosome rearrangement, or genomic modifications, such as massive accumulation of heterochromatin, or multiplication of the rDNA regions.

Interestingly, the number of the observed rDNA locations in the Barombi Mbo cichlids (14 signals per 2n; Figure 3), despite being invariable within the flock, is higher than the usual (and median) observed number for both 5S and 45S rDNA (i.e., including 28S rDNA) subunits. In ray-finned fishes, the rDNA is most commonly present as one single block (i.e., one pair of signals per 2n) for each type, i.e., 5S and 45S [74], although the observed numbers in this study (i.e., four pairs for 5S and three pairs for 28S/45S rDNA) are not outside the known range [74,75,76]. Even in African cichlids, variations in the number of rDNA gene clusters are frequently observed, and similarly, the presence of two clusters in homologous chromosomes is the most common pattern for both 5S and 28S rDNA traits [70,75]. The number of rDNA clusters per diploid genome ranges from 2 to 15 for 5S rDNA, and from 2 to 6 for 45S (28S) rDNA in cichlids [70]. The closely related Nile tilapia (*O. niloticus*) differs from the Barombi Mbo cichlids in the rDNA signals, possessing one 5S rDNA cluster less with no signal on the largest st/a chromosome pair (i.e., it has only three pairs, while Barombi Mbo cichlids have four pairs of 5S rDNA signals; [70,77], whereas the 28S rDNA signal is similar—three pairs in tilapia and Barombi cichlids). Variability in rDNA clusters from the Nile tilapia, and no observed variability within the Barombi Mbo cichlids therefore suggests that the genome of the Barombi Mbo cichlids possibly differentiated in the ancestor before the intralacustrine differentiation of the species flock, but not after.

Numbers of GC-rich regions with accumulated heterochromatin were the only analysis in which slight differences among species were observed. The highest numbers of signals (up to 18) were observed in metaphase chromosomes of *Sarotherodon linnellii* and *Konia eisentrauti*, while the lowest number (14) were observed in *Pungu maclareni*. However, we also observed substantial variability in the numbers of the GC-rich regions among individuals of the same species, and even among metaphase chromosomes within one individual. Such differences in the GC-rich scoring could be caused by different condensation of chromosomes and may, therefore, represent an artefactual observation. Like with rDNA, observation of multiple GC-rich regions is rare among genomes of teleost fishes, and has been observed in only a limited number of studies e.g., [78]. Since rDNA sites in eukaryotes are generally known as regions of substantial GC enrichment [79], multiple accumulation of GC-rich regions observed in this study may be simply associated with the multiple (14) repetitive rDNA signals observed in the cichlid chromosomes by applying FISH staining.

In conclusion, we integrated various comparative cytogenetic approaches, and we present the pilot cytogenetic study of the endemic Barombi Mbo cichlid species flock. Our results show karyotypic and chromosomal stability in these species, which have undergone rapid sympatric ecological speciation. Our results suggest that inter-chromosomal rearrangements followed by the karyotype incompatibility have likely not contributed to the speciation processes in the lake. Nevertheless, further advanced molecular cytogenetic techniques, such as chromosome painting or CGH (comparative genomic hybridization), will be required to verify detailed synteny of the chromosomal pairs across species, or more subtle differences undetectable by the approaches applied in this study. We finally consider our study as complementary evidence for any future molecular genomic studies focused on cichlids from the Barombi Mbo crater lake.

## 4. Materials and Methods

### 4.1. Specimens

We examined 29 individuals from 11 species as described in Table 1. Samples were collected and the research was conducted under research permits (numbers: 0000047,49/MINRESI/B00/C00/ C10/nye, 0000116,117/MINRESI/B00/C00/C10/C14, 000002-3/MINRESI/B00/C00/C10/C11, 0000032,48 -50/MINRESI/B00/C00/C10/C12) issued by the Ministry of Scientific Research and Innovation in Cameroon. Valid Animal Use Protocol issued by Ministry of Agriculture, Czech Republic (No. CZ 02386, approved 25th September 2014) was in force during this study in IAPG. Further information about numbers of examined cells are listed in Table 1. Due to the limited access to the studied material, we did not perform advanced stainings on chromosomes of *S. caroli* and *K. dikume*. In these species, only Giemsa-stained karyotypes are presented in this study.

### 4.2. Chromosome Preparation and Giemsa Staining, CMA_3_ Staining, and C-Banding

Metaphase chromosomes were prepared according to [80] with slight modifications. Briefly, fish were injected with 0.1% colchicine solution (1 mL/100 g of body weight) and euthanized after 45 min using an overdose of anaesthetic (phaenoxyethanol). The kidneys were dissected in 0.075 M KCl at room temperature. The cell suspension was hypotonized for 30 min in 0.075 M KCl, fixed in freshly prepared fixative (methanol: acetic acid 3:1, *v*/*v*), washed twice in fixative and spread onto slides. Alternatively, the chromosomes of the rare species, *K. dikume*, were obtained from the fin regenerate according to the protocol of [81], previously successfully applied to cichlids [55]. Chromosomal spreads were stained with Giemsa solution (5%, 10 min) to identify the number and morphology of chromosomes in all 11 species used in this study. To visualize the blocks of constitutive heterochromatin, the C-banding staining was performed according to [82], with slight modifications as described in [83]. After C-banding, the chromosomes were counterstained with Vectashield DAPI anti-fade medium (Vector Laboratories, Burlingame, CA, USA) to enhance the contrast, and the microphotographs were taken in the fluorescent regime and inverted. To reveal the GC genome composition, Chromomycin A_3_ (CMA_3_) staining was performed as described by [84] using Vectashield DAPI anti-fade medium as a mounting reagent (Vector Laboratories, Burlingame, CA, USA).

### 4.3. Fluorescence in situ Hybridization with Telomeric Probe and rDNA Genes

FISH with Cy3-labelled telomeric PNA probe was performed according to the manufacturer’s instructions (Telomere PNA FISH Kit/Cy3, Dako, Denmark). Probes for rDNA FISH experiments were produced by PCR with the primer pairs as follows: (i) 28S rDNA: 5’-AAACTCTGGTGGAGGTCCGT-3’ and 5’-CTTACCAAAAGTGGCCCACTA-3’ [85]; (ii) 5S rDNA: 5’-TACGCCCGATCTCGTCCGATC-3’ and 5’-CAGGCTGGTATGGCCGTAAGC-3’ [86]. The PCR reactions were carried out as described in [87]. Probes were indirectly labelled with biotin-16-dUTP (Roche, Mannheim, Germany) and digoxigenin-11-dUTP (Roche) through PCR reamplification of PCR products. Reamplification was carried out under the same conditions as the previous PCR reaction. A hybridization mixture was made, consisting of labelled and precipitated PCR products of both genes, hybridization buffer [88], and salmon sperm blocking DNA (15 μg/slide; Sigma-Aldrich, St. Louis, MO, USA). The hybridization and detection procedures were carried out under conditions described by [88]. The biotin-dUTP-labelled probes were detected by Invitrogen Cy™3-Streptavidin (Invitrogen, San Diego, CA, USA; cat. no. 43-4315), the digoxigenin-dUTP-labeled probes were detected by anti-digoxigenin-rhodamine (cat. no. 11207750910). The slides were mounted with Vectashield DAPI anti-fade medium (Vector Laoratories, Burlingame, CA, USA).

### 4.4. Microscopy and Image Processing

Chromosomal preparations were examined by a ZEISS Axio Imager.Z2 epifluorescence microscope. Images of metaphase chromosomes were recorded with a CoolCube 1 camera (MetaSystems, Altlussheim, Germany). Analyses of images were performed in the IKAROS and ISIS imaging programs (MetaSystems, Altlussheim, Germany). The captured digital images from FISH experiments were pseudocolored and superimposed using Adobe Photoshop software, version CS5. For CMA_3_/DAPI staining, the CMA_3_ signal was inserted into the red and the DAPI signal into the green channel to enhance the contrast between these two types of signals. In karyotypes, chromosomes were ordered in decreasing size and the chromosomal categories were classified according to Levan et al. [89].

## Figures and Tables

**Figure 1 ijms-20-04994-f001:**
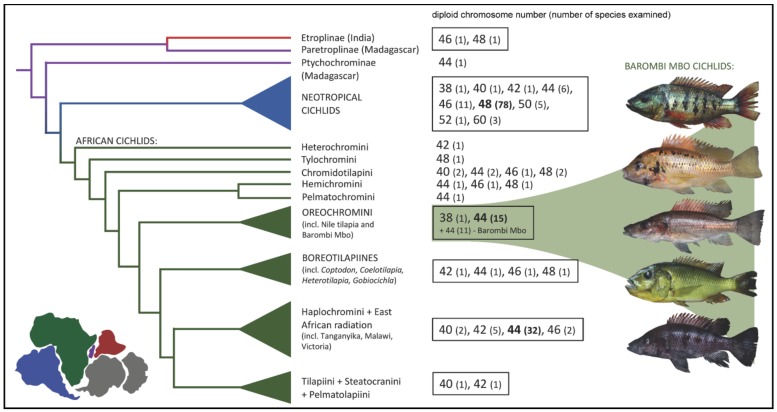
Schematic evolution of cichlid karyotypes. Data reviewed from [22,23,52,53,54,55,56,57]. Modal chromosome numbers for African (2n = 44) and Neotropical (2n = 48) cichlids are highlighted in bold. Phylogenetic relationships after [38,39]. Note that Barombi Mbo cichlids belong in the Oreochromini tribe, and the phylogenetic position of this tribe among other African cichlids shows that they are only distantly related to the cichlids from the Tanganyika, Malawi and Victoria lakes. Photos show *Konia dikume*, *Pungu maclareni*, *Stomatepia mongo*, *Sarotherodon linnellii* and *St. pindu*.

**Figure 2 ijms-20-04994-f002:**
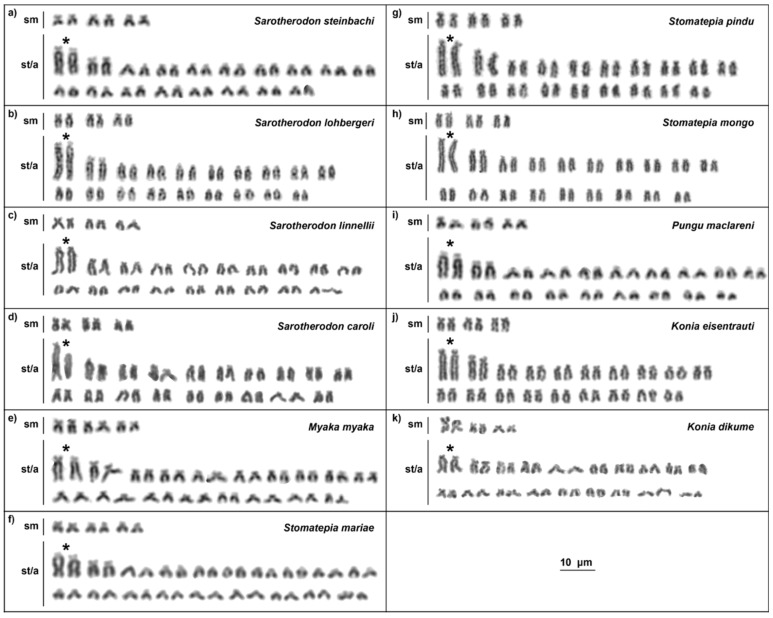
Karyotypes of all 11 cichlids from Barombi Mbo crater lake arranged from Giemsa-stained chromosomes. All 11 species from the lake possess an identical karyotype with 2n = 44 (NF = 50). (**a**) *Sarotherodon steinbachi*, (**b**) *S. lohbergeri*, (**c**) *S. linnellii*, (**d**) *S. caroli*, (**e**) *Myaka myaka*, (**f**) *Stomatepia mariae*, (**g**) *St. pindu*, (**h**) *St. mongo*, (**i**) *Pungu maclareni*, (**j**) *Konia eisentrauti*, and (**k**) *K. dikume*; sm, submetacentric and st/a, subtelocentric-acrocentric chromosomes. The enlarged subtelocentric chromosome pair, which is also characteristic of the Nile tilapia (*Oreochromis niloticus*) karyotype, is marked with an asterisk.

**Figure 3 ijms-20-04994-f003:**
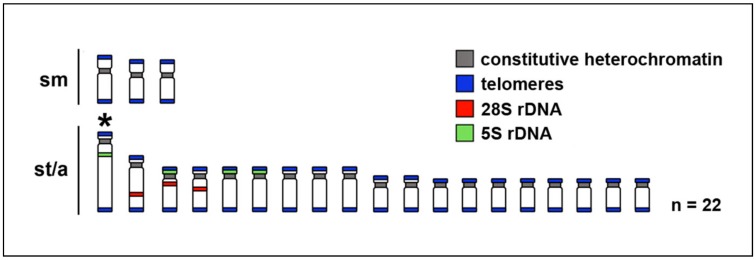
Ideogram (schematic representation of the haploid chromosome set) of the Barombi Mbo cichlid fishes after cytogenetic protocols: C-banded chromosomal regions are marked by grey color; telomeric signals after PNA-telomeric FISH are marked by blue color, and rDNA signals are marked by red (28S rDNA) and green (5S rDNA). See Figure 4 and Figure 5 for the actual results of the methods visualized in the ideogram; abbreviations: sm, submetacentric and st/a, subtelocentric-acrocentric chromosomes. The enlarged subtelocentric chromosome pair, which is also characteristic of the Nile tilapia (*Oreochromis niloticus*) karyotype, is marked with an asterisk.

**Figure 4 ijms-20-04994-f004:**
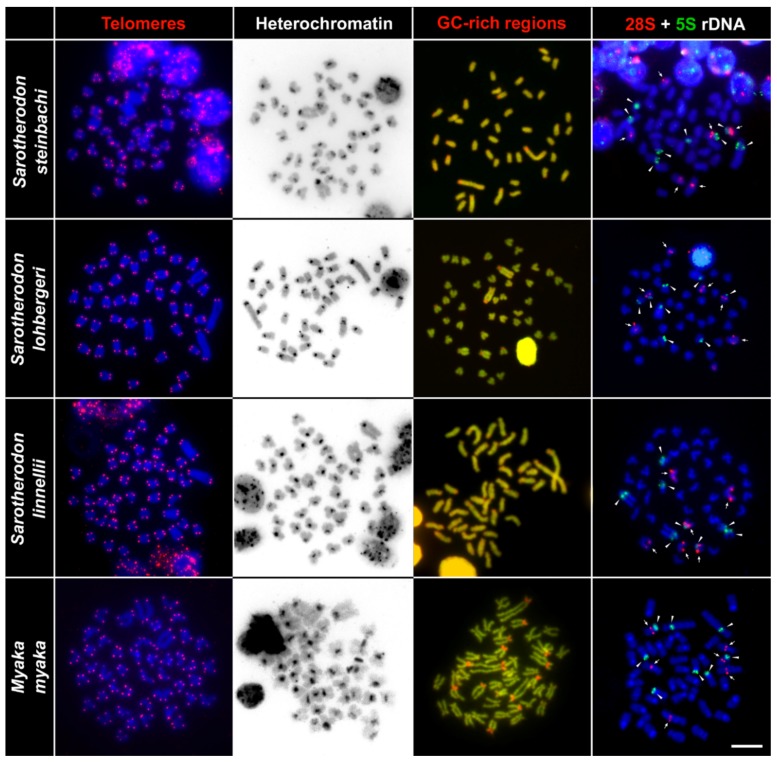
Comparative chromosome analyses of *Sarotherodon steinbachi*, *S. lohbergeri*, *S. linnellii*, and *Myaka myaka*. First column: DAPI-stained chromosomes (blue) with telomere repeat hybridization signals (red) located only in the telomeric region assuming no chromosome rearrangements. Second column: inverted DAPI-stained C-banding pattern highlights clusters with constitutive heterochromatin located in the pericentromeric regions. Third column: DAPI-stained metaphase chromosomes (green) with signals of GC-rich sites (red). Fourth column: DAPI stained metaphase chromosomes (blue), with six 28S rDNA (red, highlighted by arrows), and eight 5S rDNA (green, highlighted by arrowheads) hybridization signals in each species. Bar equals 10 µm.

**Figure 5 ijms-20-04994-f005:**
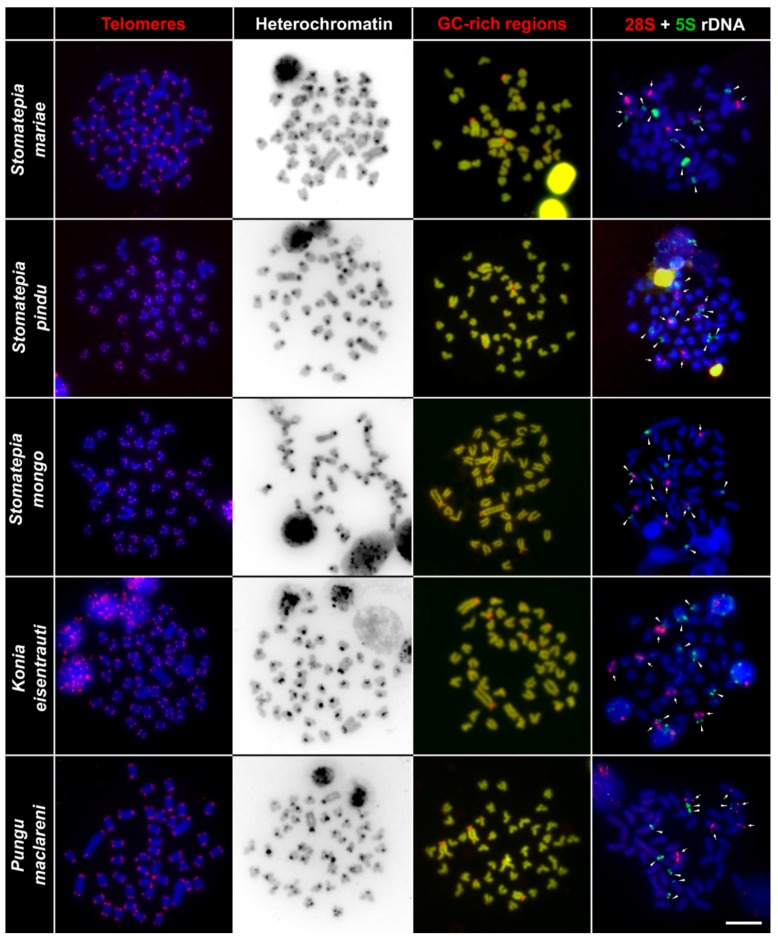
Comparative chromosome analyses of *Stomatepia mariae*, *St. pindu*, *St. mongo*, *Konia eisentrauti*, and *Pungu maclareni*. First column: DAPI-stained chromosomes (blue) with telomere repeat hybridization signals (red) located only in the telomeric region assuming no chromosome rearrangements. Second column: inverted DAPI-stained C-banding pattern highlights clusters with constitutive heterochromatin located in the pericentromeric regions. Third column: DAPI-stained metaphase chromosomes (green), signals of GC-rich sites (red). Fourth column: DAPI stained metaphase chromosomes (blue), with six 28S rDNA (red, highlighted by arrows), and eight 5S rDNA (green, highlighted by arrowheads) hybridization signals in each species. Bar equals 10 µm.

**Table 1 ijms-20-04994-t001:** Numbers of individuals and cells analyzed in this study.

Species	No. of Individuals	No. of Cells Examined				
		Giemsa	C-Banding	CMA_3_	Telomeres	FISH
*Konia eisentrauti*	1♂, 2♀	35	30	30	30	15
*Konia dikume*	1	8	n/a	n/a	n/a	n/a
*Myaka myaka*	4♂	35	30	30	30	16
*Pungu maclareni*	1♀	35	30	30	30	15
*Sarotherodon steinbachi*	3 juveniles	60	30	30	30	12
*Sarotherodon lohbergeri*	2 juveniles	35	30	30	30	15
*Sarotherodon linnellii*	2♀, 1 juveniles	45	30	30	30	18
*Sarotherodon caroli*	2	10	n/a	n/a	n/a	n/a
*Stomatepia mariae*	4 juveniles	35	30	30	30	10
*Stomatepia pindu*	3 juveniles	40	30	30	30	13
*Stomatepia mongo*	3 juveniles	30	30	30	30	12

♂: male individual; ♀: female individual.

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
