# Peer review of "Adaptive Radiation from a Chromosomal Perspective: Evidence of Chromosome Set Stability in Cichlid Fishes (Cichlidae: Teleostei) from the Barombi Mbo Lake, Cameroon"

_ijms, 2019, doi:10.3390/ijms20204994_

Round 1

Reviewer 1 Report

In their work Majtánová et al. performed a very comprehensive cytogenetic analysis of 11 crater lake cichlids from Barombi Mbo, Cameroon. As these species compose an adaptive radiation that likely arose in sympatry, it is a very important and relevant hypothesis if differences in chromosome number or more generally cytogenetic variation contributed to this process. In their analysis they use a combination of classical methods like Giemsa staining, but also fluorescent in situ hybridization to be able to also pinpoint variation on a more micro-chromosomal levels. The main conclusion of their work is that karyotypes and chromosomal structure are overall conserved and stable across the Barombi Mbo radiation. And while it would have been extremely exciting to see variation that possibly contributed to sympatric speciation, it was an important and very relevant hypothesis to test. The work is very well introduced, presented, structured, discussed and the quality of the data appears very high (I say appears, because I am not an expert on cytogenetic analysis). Its a significant, scientifically sound and in the way how this story was framed also original and novel work. I therefore recommend acceptance of this work with a few minor comments that I would like to ask the authors to incorporate or comment on:

1. Line 21ff: As two genera are abbreviated with “S.” it is hard to tell them apart throughout the text. I would suggest to use “St.” for Stomatepia.
2. Line 28: Although the more specialist reader will be aware what m/sm and st/a means,  I would suggest to avoid the abbreviations in the abstract to make it a bit more accessible to the general reader.
3. Line 30 :Although I read the last sentence of the abstract “The results suggested that similar traits of genome evolution occurred during chromosome evolution in this group.” several times, its not entirely clear to me what is meant/concluded here (especially what “similar traits of genome evolution” are). Maybe the authors can clarify.
4. Recent work (mainly by the Kocher lab, e.g.: https://www.ncbi.nlm.nih.gov/pubmed/30563180) reported on B chromosomes in East African cichlids. Is that something the authors also observed in any of their Lake Barombi Mbo species or that the authors think to be worth discussing?
5. Although the authors very nicely and comprehensively put their work in context to previous studies (the authors state that there are 130 cichlid species for which karyotypes have been analyzed), I was wondering if a very simplified cichlid phylogeny where information about chromosome number evolution in the most important genera and including the Lake Barombi Mbo radiation is synthesized would help non-cichlid researchers.

Author Response

Dear reviewer,

Thank you very much for your review! We changed our manuscript according to your suggestions. Please see our detailed comments below.

Yours sincerely,

Zuzana Majtánová and co-authors

Comments and Suggestions for Authors

In their work Majtánová et al. performed a very comprehensive cytogenetic analysis of 11 crater lake cichlids from Barombi Mbo, Cameroon. As these species compose an adaptive radiation that likely arose in sympatry, it is a very important and relevant hypothesis if differences in chromosome number or more generally cytogenetic variation contributed to this process. In their analysis they use a combination of classical methods like Giemsa staining, but also fluorescent in situ hybridization to be able to also pinpoint variation on a more micro-chromosomal levels. The main conclusion of their work is that karyotypes and chromosomal structure are overall conserved and stable across the Barombi Mbo radiation. And while it would have been extremely exciting to see variation that possibly contributed to sympatric speciation, it was an important and very relevant hypothesis to test. The work is very well introduced, presented, structured, discussed and the quality of the data appears very high (I say appears, because I am not an expert on cytogenetic analysis). Its a significant, scientifically sound and in the way how this story was framed also original and novel work. I therefore recommend acceptance of this work with a few minor comments that I would like to ask the authors to incorporate or comment on:

Line 21ff: As two genera are abbreviated with “S.” it is hard to tell them apart throughout the text. I would suggest to use “St.” for Stomatepia.

- corrected

Line 28: Although the more specialist reader will be aware what m/sm and st/a means, I would suggest to avoid the abbreviations in the abstract to make it a bit more accessible to the general reader.

- corrected

Line 30 :Although I read the last sentence of the abstract “The results suggested that similar traits of genome evolution occurred during chromosome evolution in this group.” several times, its not entirely clear to me what is meant/concluded here (especially what “similar traits of genome evolution” are). Maybe the authors can clarify.

- Thank you for this remark. We have changed the sentence to: “The results suggest the evolutionary stability of chromosomal set, therefore, the large-scale chromosomal rearrangements seem to be unlikely associated with the sympatric speciation in Barombi Mbo.”

Recent work (mainly by the Kocher lab, e.g.: https://www.ncbi.nlm.nih.gov/pubmed/30563180) reported on B chromosomes in East African cichlids. Is that something the authors also observed in any of their Lake Barombi Mbo species or that the authors think to be worth discussing?

- We have not observed any B chromosomes among species from Barombi Mbo. Nevertheless, we added the information about identified B chromosomes among African cichlid species into the discussion (lines 269-270).

Although the authors very nicely and comprehensively put their work in context to previous studies (the authors state that there are 130 cichlid species for which karyotypes have been analyzed), I was wondering if a very simplified cichlid phylogeny where information about chromosome number evolution in the most important genera and including the Lake Barombi Mbo radiation is synthesized would help non-cichlid researchers.

- Thank you for this comment. We have added a figure summarizing the karyotype evolution in cichlids and highlighting the phylogenetic position of the Barombi Mbo clade.

Reviewer 2 Report

     Lake Barombi Mbo in Cameroon has a rich endemic cichlid species flock that presumably is the result of adaptive radiation over a relatively short time period. One approach to gain insight into this systesm is to assess whether a chromosomal mode of speciation drove rapid radiation. Majtanova et al. employed four chromosome staining methods to examine metaphase chromosome spreads for the 11 cichlids of Barombi Mbo. Surprisingly, the found no chromosomal differentiation, which then does not support the chromosomal speciation explanation, and rather supports an ecological mechanism for adaptive radiation. The choice of methods was appropriate and the results were about as clear as can be expected of karyotypic studies. The presentation can be sharpened up a bit. I make a few comments here and also provide a marked manuscript to guide revision.      

     Abstract. – Many readers will not know shorthand descriptors of chromosome structure, so m/sm and st-/a should be spelled out in the introduction.

     Discussion. – At line 251, it is not clear to me what CMA refers to

     At line 294, it is species, not individuals that are referred to.

     Methods. At line 304, so that others can build on this work the city where Dako is located should be given.

     References. – I marked many small breaks from journal citation stylistics on the manuscript document.

Author Response

Dear reviewer,

We would like to kindly thank you for your great job with the review. We really appreciate your effort in detailed focus on our manuscript which helped us to improve it. Thank you for providing the detailed pdf with suggested changes. We have followed your suggestions and incorporated most of the changes in the manuscript.

We provide detailed responses to your comments below.

Thank you again for the helpful review.

Yours sincerely,

Zuzana Majtánová and co-authors

Comments and Suggestions for Authors

     Lake Barombi Mbo in Cameroon has a rich endemic cichlid species flock that presumably is the result of adaptive radiation over a relatively short time period. One approach to gain insight into this systesm is to assess whether a chromosomal mode of speciation drove rapid radiation. Majtanova et al. employed four chromosome staining methods to examine metaphase chromosome spreads for the 11 cichlids of Barombi Mbo. Surprisingly, the found no chromosomal differentiation, which then does not support the chromosomal speciation explanation, and rather supports an ecological mechanism for adaptive radiation. The choice of methods was appropriate and the results were about as clear as can be expected of karyotypic studies. The presentation can be sharpened up a bit. I make a few comments here and also provide a marked manuscript to guide revision.

Abstract. – Many readers will not know shorthand descriptors of chromosome structure, so m/sm and st-/a should be spelled out in the introduction.

- corrected

     Discussion. – At line 251, it is not clear to me what CMA refers to

- We have corrected the sentence to “Numbers of GC-rich regions with accumulated heterochromatin were the only analysis in which differences among species were observed.“

     At line 294, it is species, not individuals that are referred to.

- corrected

     Methods. At line 304, so that others can build on this work the city where Dako is located should be given.

- thank you for this remark, we have added the missing information

     References. – I marked many small breaks from journal citation stylistics on the manuscript document.

- thank you again, we have made an extensive correction of references.